# A visualization reporter system for characterizing antibiotic biosynthetic gene clusters expression with high-sensitivity

Xiang Liu [1,2,3], Jine Li[1,3], Yue Li[1], Junyue Li[1,2], Huiying Sun[1,2], Jiazhen Zheng[1,2], Jihui Zhang [1✉] & Huarong Tan [1,2✉]

The crisis of antibiotic resistance has become an impending global problem. Genome sequencing reveals that streptomycetes have the potential to produce many more bioactive compounds that may combat the emerging pathogens. The existing challenge is to devise sensitive reporter systems for mining valuable antibiotics. Here, we report a visualization reporter system based on Gram-negative bacterial acyl-homoserine lactone quorum-sensing (VRS-bAHL). AHL synthase gene (*cviI*) of *Chromobacterium violaceum* as reporter gene is expressed in Gram-positive *Streptomyces* to synthesize AHL, which is detected with CV026, an AHL deficient mutant of *C. violaceum*, via its violacein production upon AHL induction. Validation assays prove that VRS-bAHL can be widely used for characterizing gene expression in *Streptomyces*. With the guidance of VRS-bAHL, a novel oxazolomycin derivative is discovered to the best of our knowledge. The results demonstrate that VRS-bAHL is a powerful tool for advancing genetic regulation studies and discovering valuable active metabolites in microorganisms.

[1] State Key Laboratory of Microbial Resources, Institute of Microbiology, Chinese Academy of Sciences, Beijing, China. [2] College of Life Sciences, University of Chinese Academy of Sciences, Beijing, China. [3] These authors contributed equally: Xiang Liu, Jine Li. ✉email: zhang.jihui@im.ac.cn; tanhr@im.ac.cn

Microorganisms are the most abundant source of natural antibiotics for clinical application, and have the potential to produce many more secondary metabolites than previously thought, which might be used as drug leads[1–3]. However, the biosyntheses of these compounds are under stringent control of intricate regulatory networks[4,5]. Various genetic tools have been invented for elucidating the regulatory mechanisms and biosynthetic pathways, and played considerable roles in unearthing the valuable bioactive metabolites from microorganisms, especially streptomycetes[6,7].

Development of reporter systems has revolutionized the way of characterizing gene expression[6]. Antibiotic resistance genes are conventional reporters and proved useful in many *Streptomyces* species, but they are not ideal for quantitative determination of gene expression[8,9]. Green fluorescent protein (GFP) and luciferase coding genes serving as reporters showed advantages because the signal could be instantly monitored and quantified with luminescence detecting instruments[10–13]. Alternatively, chromogenic reporter genes, such as *xylE* (catechol 2,3-dioxygenase coding gene), *gusA* (β-glucuronidase coding gene), *idgS* (indigoidine synthetase gene) and *whiE* (grey-brown spore pigment synthetic genes), were chosen to construct reporter systems for rapid assessment of gene promoter activity[14–17]. However, the efficiency of fluorescent and chromogenic gene reporter systems is often dramatically interfered by the pigments from *Streptomyces*[7]. Thus, a sensitive reporter system with eliminated background interference is still an urgent need and worth developing.

Quorum sensing (QS) is a cell–cell communication process and well-known for its hypersensitivity[18]. Specific QS signaling systems are employed in Gram-positive and Gram-negative bacteria, respectively, to coordinate population behavior by producing and responding to signaling molecules[19]. For example, γ-butyrolactones or butenolides are dominant signaling molecules in Gram-positive *Streptomyces* and broadly involved in the regulation of antibiotic production or morphological differentiation[20]. Acyl-homoserine lactone (AHL) QS systems, typically consisting of AHL signaling molecules and their cognate receptors are generally exclusive to Gram-negative bacteria. For example, C6-HSL is a short-chain AHL signaling molecule as the ligand of receptor CviR in *Chromobacterium violaceum* CV31532. The AHL-receptor complex would activate the transcription of violacein biosynthetic genes in *C. violaceum* and subsequently the production of violacein, an intracellular purple pigment for visible observation. CV026 is an AHL and violacein deficient mutant of *C. violaceum* CV31532, in which the AHL synthase gene *cviI* and the violacein biosynthetic repressor gene *vioS* were disrupted via transposon-mediated mutagenesis. The response of CviR to AHL signaling molecules was so sensitive that nanomolar concentration of exogenously added C6-HSL could restore violacein production in CV026[21–23]. Overall, microbial QS systems widely participate in many physiological processes, such as biofilm formation, exoprotease synthesis and antibiotic production[24,25]. In addition to mediating intraspecies group behaviors, QS systems could also interact with others, resulting in interspecies communications[26]. Moreover, QS signaling processes could be quite complicated if external factors, such as QS antagonists or agonists, are involved[27,28]. Therefore, QS signaling systems serve as an alternative bio-element resource for developing novel manipulation or evaluation techniques of gene expression and regulation[29–32].

In this study, we aim to construct a sensitive visualization reporter system based on Gram-negative bacterial AHL QS (VRS-bAHL) for Gram-positive bacteria to determine gene expression via AHL production in *Streptomyces*. Meanwhile, since no evidence of *Streptomyces* producing AHLs has been reported up to now, clean background of the detection with this strategy is expected. Thus, this system would facilitate the discovery of new antibiotics and yield improvement of existing ones.

## Results

### Construction and evaluation of VRS-bAHL in *Streptomyces* species.
Despite the rapid development of various gene reporter systems, the interference of secondary metabolites produced by host cells and low sensitivity have been serious concerns. To solve these problems, a visualization reporter system based on Gram-negative bacterial AHL QS (VRS-bAHL) was constructed in representative bacteria, streptomycetes. This strategy uses AHL synthase gene *cviI* as the reporter gene driven by target promoters of *Streptomyces*. Once *cviI* is expressed, its coding product, CviI, would catalyze the synthesis of C6-HSL using S-adenosyl-L-methionine (SAM) and acyl-acyl carrier protein (acyl-ACP) as substrates[33]. C6-HSL is diffusible from *Streptomyces* cells to surroundings and can be recognized by its cognate receptor CviR of the indicator strain (CV026), leading to the biosynthesis of violacein, a visible purple pigment (Fig. 1a).

To test the feasibility of VRS-bAHL in *Streptomyces*, *cviI* was expressed in seven different *Streptomyces* strains (*S. ansochromogenes*, *S. longshengensis*, *S. virginiae*, *S. griseus*, *S. venezuelae*, *S. coelicolor* and *S. lividans*) under the control of constitutive *hrdB* promoter ($P_{hrdB}$) from *S. coelicolor*[34], resulting in recombinant *Streptomyces* strains (OEcviIs). Then, double-layer plate method (as described in Methods) was used to detect AHL production. As expected, the expression of *cviI* in OEcviIs grown on MS agar medium was clearly indicated by the production of purple violacein in CV026, which was not present in the corresponding negative control strains (NCs) harboring the blank plasmid (Fig. 1b). Since metabolites produced in *Streptomyces* and *cviI* expression may vary in different media, which could potentially interfere with the performance of VRS-bAHL, this system was further tested on MYM, TSB and YEME agar media. Encouragingly, similar results were obtained for OEcviIs as on MS despite some variations in zone size and intensity of the purple pigment. It was also noteworthy that no obvious background interference was seen in all the tests (Fig. 1b). Furthermore, considering the diverse culturing conditions of *Streptomyces*, more AHL detection methods (agar plug method, co-cultivation method, Oxford cup method for liquid fermentation broth or extracts and another quick detection approach using *E. coli* reporter strain[35]) were investigated. The results confirmed their efficiency in coupling with VRS-bAHL (Supplementary Figs. 1 and 2). Moreover, we found that 10 nM of C6-HSL could induce the violacein production of CV026 (Supplementary Fig. 3), suggesting that VRS-bAHL is a sensitive approach for detecting gene expression. While, owing to visible pigment of violacein, quantitative detection can be done by directly measuring and comparing the violacein zone area or determining the optical density of violacein extracted with dimethyl sulphoxide (DMSO) (Supplementary Fig. 3). Hence, VRS-bAHL was demonstrated to be compatible with *Streptomyces* under different culture conditions with high sensitivity and efficiency.

### Characterization of genetic regulation of *ovm* gene cluster guided by VRS-bAHL.
Elucidation of regulatory mechanisms is a basis for discovering natural products from microbial resources, which relies on convenient and precise assessment of gene or gene cluster expression. To investigate whether VRS-bAHL can be used in characterizing genetic regulation of *Streptomyces*, we established a testing system. Our previous study based on *gusA* reporter system in *S. coelicolor* M1146 showed that OvmZ and OvmW as a pair of regulatory proteins could coordinately activate the key structural gene

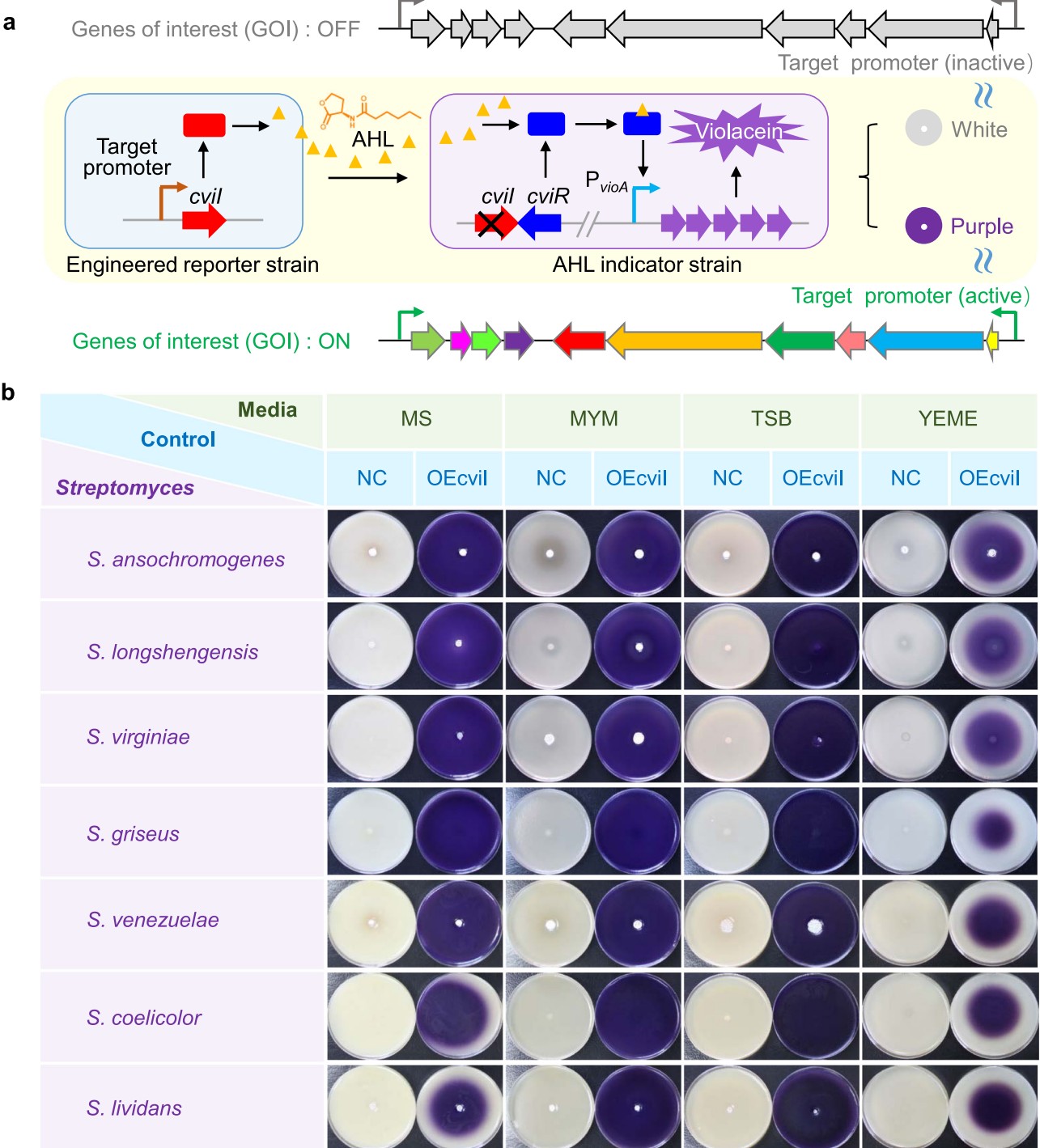

**Fig. 1 Schematic illustration and feasibility verification of VRS-bAHL in *Streptomyces*. a** Schematic illustration of VRS-bAHL. VRS-bAHL was constructed based on AHL QS of *C. violaceum* CV31532. Specifically, *cviI* (the AHL synthase gene of CV31532) was used as the reporter gene in *Streptomyces* and CV026 (the *cviI* deficient mutant of CV31532) as AHL indicator strain. *cviI* was fused with target promoters and then introduced into *Streptomyces*. The resulting AHL signaling molecules, such as C6-HSL, can be recognized by receptor CviR in indicator strain CV026 to form a complex, whose binding to $P_{vioA}$, the promoter of violacein biosynthetic genes, would activate violacein production. Purple pigment (violacein) can be directly observed or measured for quantification using suitable instruments, which indicates the expression level of genes of interest (GOI). **b** Feasibility verification of VRS-bAHL in *Streptomyces*. OEcviI (positive control strain harboring pSPhrdB-cviI for overexpression of *cviI* driven by $P_{hrdB}$ on pSET152) and NC (negative control strain harboring the blank plasmid pSET152) strains were inoculated at the center of agar media plates. After incubation for three days, double-layer plate method was applied to detect AHL via violacein formation in indicator strain CV026. Seven *Streptomyces* strains (*S. ansochromogenes* 7100, *S. longshengensis* CGMCC 4.1101, *S. virginiae* CGMCC 4.1530, *S. griseus* IFO 13350, *S. venezuelae* ATCC 10712, *S. coelicolor* A3(2) and *S. lividans* TK23) were chosen for the feasibility verification of VRS-bAHL on four types of media (MS, MYM, TSB and YEME), and two independent experiments displayed similar results.

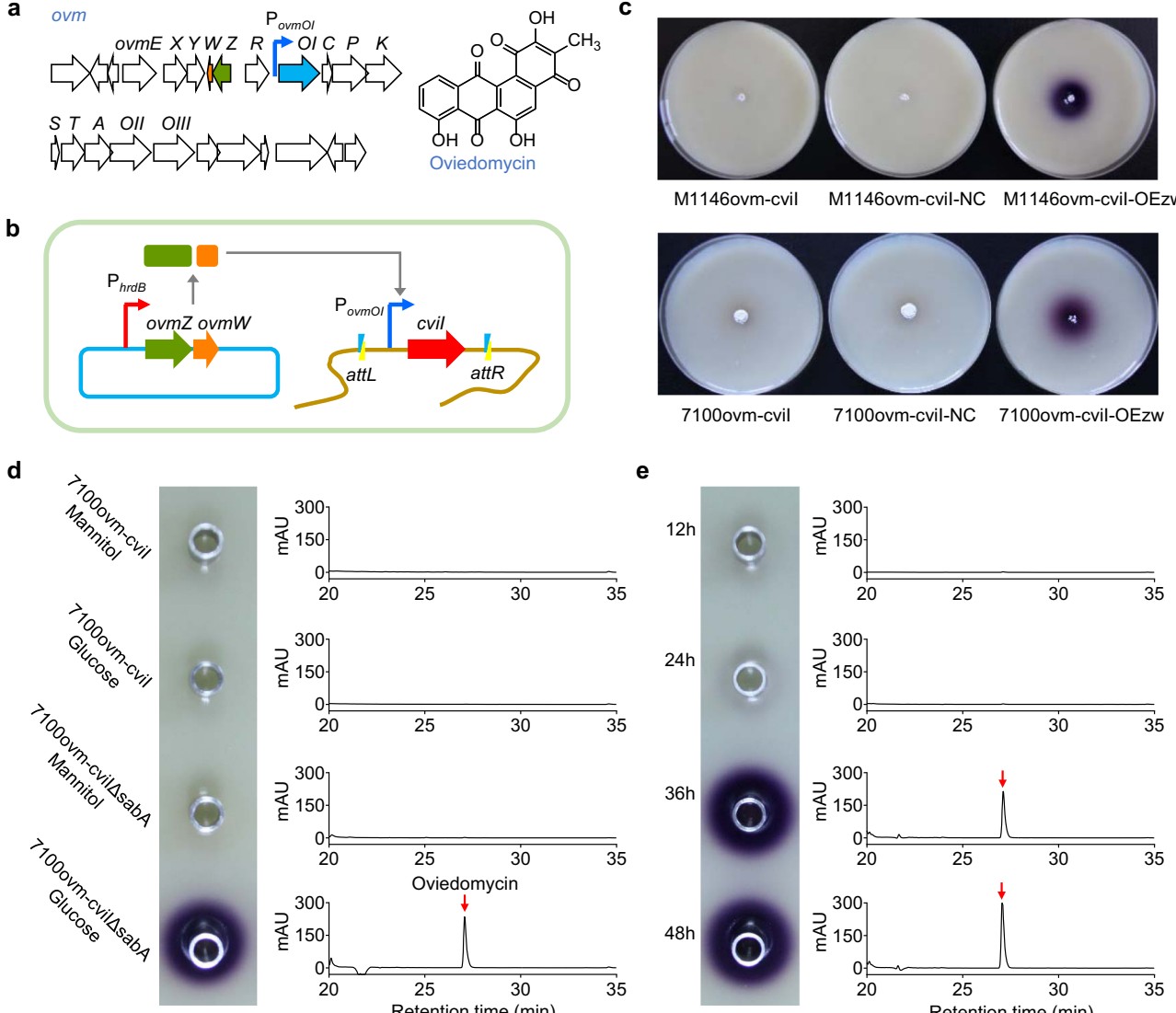

**Fig. 2 Characterization of genetic regulation of *ovm* gene cluster guided by VRS-bAHL. a** Genetic organization of oviedomycin biosynthetic gene cluster (*ovm*) in *S. ansochromogenes* 7100 and its chemical structure. **b** Schematic illustration of VRS-bAHL in detecting the activation of *ovmOI* promoter (P*ovmOI*) by OvmZ and OvmW. The workflow is as follows: *cviI* is driven by P*ovmOI*, while *ovmZ* and *ovmW* are driven by the constitutive promoter P*hrdB* from *S. coelicolor*. Once P*ovmOI* is activated by OvmZ and OvmW, *cviI* can be expressed and the synthesized C6-HSL would induce violacein production in CV026. **c** Characterization of OvmZ and OvmW regulating *ovmOI* with VRS-bAHL in *S. coelicolor* M1146 and *S. ansochromogenes* 7100. M1146ovm-cviI, a reporter strain derived from *S. coelicolor* M1146 containing *cviI* expression plasmid driven by P*ovmOI* on plasmid pIJ10500K; M1146ovm-cviI-NC, derivative strain of M1146ovm-cviI containing blank plasmid pKC1139 as negative control strain; M1146ovm-cviI-OEzw, derivative strain of M1146ovm-cviI containing *ovmZ* and *ovmW* co-overexpression plasmid pKC1139::P*hrdB*ZW. Similarly, the strains derived from *S. ansochromogenes* 7100 were constructed. All *Streptomyces* strains were cultivated on MS agar medium. Double-layer plate method was applied to detect AHL in the tests. **d** Characterization of *ovm* activation in 7100ovm-cviIΔsabA with VRS-bAHL. **e** Determination of *ovm* activation time in 7100ovm-cviIΔsabA with VRS-bAHL. In both **d**, **e**, AHL was detected with CV026 using Oxford cup method and the oviedomycin production (red arrow) was analyzed by HPLC as previously described[37]. Data shown in **c** to **e** are representative of two independent experiments that displayed similar results.

promoter of *ovmOI* (P*ovmOI*) in *ovm*, the oviedomycin biosynthetic gene cluster (BGC)[36] (Fig. 2a). In this study, we employed VRS-bAHL to characterize the activation of P*ovmOI* mediated by OvmZ and OvmW (Fig. 2b). As expected, violacein production in CV026 was induced by the *ovmZ* and *ovmW* co-overexpression reporter strain (M1146ovm-cviI-OEzw), but not by the negative control strain (M1146ovm-cviI-NC). Similar results were obtained with the native oviedomycin producer, *S. ansochromogenes* 7100, confirming that OvmZ and OvmW positively regulate P*ovmOI* (Fig. 2c).

To explore the application of VRS-bAHL for those gene clusters with more complicated regulation, another *ovm*

activation approach involving nutrients and signaling molecules was tested in liquid fermentation media and the activation time was determined. Our previous study proved that butenolide signaling molecule deficiency synergized with glucose addition also can activate *ovm* gene cluster[37]. Here, we carried out the verification of this regulation relationship using VRS-bAHL. As expected, 7100ovm-cviIΔsabA (a butenolide synthase gene disruption mutant of 7100ovm-cviI) fermented in liquid glucose-SP medium successfully activated violacein production in CV026, and oviedomycin in this mutant was detected by HPLC simultaneously (Fig. 2d). Subsequently, the expression

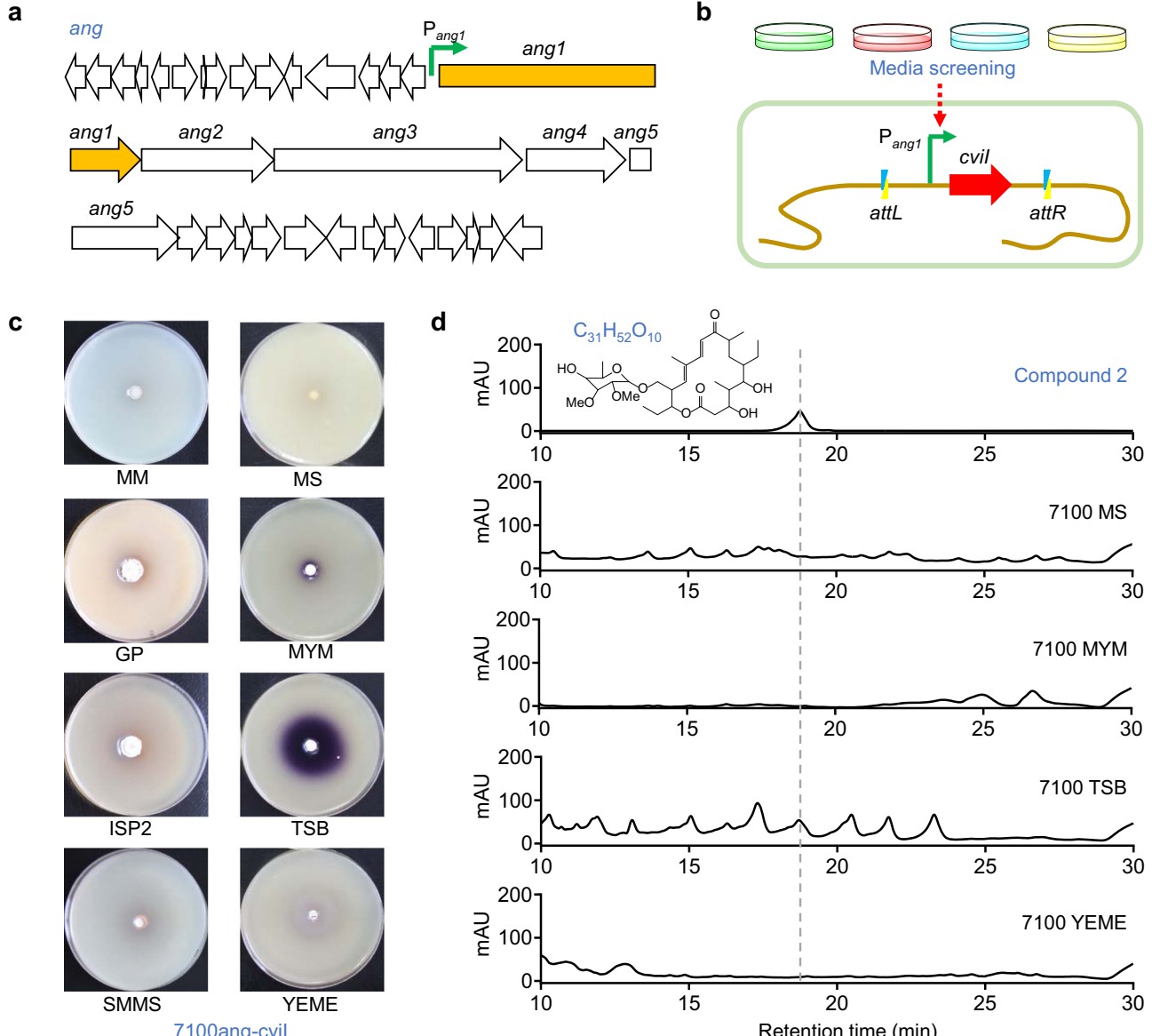

**Fig. 3 VRS-bAHL-guided identification of the activated *ang* gene cluster. a** Genetic organization of *ang* gene cluster in *S. ansochromogenes* 7100. **b** Schematic illustration of VRS-bAHL-guided identification of activated *ang* gene cluster through culture media screening. **c** Induction of violacein production in CV026 by reporter strain 7100ang-cviI (*S. ansochromogenes* 7100 harboring pPang1-cviI with *cviI* driven by P*ang1*, a promoter of *ang1* in *ang* gene cluster, on pIJ10500K) grown on eight types of agar media (MM, GP, ISP2, SMMS, MS, MYM, TSB and YEME). Double-layer plate method was performed to detect AHL production in 7100ang-cviI. The purple pigment area and intensity indicate the expression level of the reporter gene (*cviI*). **d** HPLC analysis of tylosin analogs in *S. ansochromogenes* 7100 cultured on some of the representative media. Compound 2 standard was used as reference. Dashed line indicates the peak of Compound 2. Data shown in **c** and **d** are representative of two independent experiments that displayed similar results.

initiation time of *ovm* gene cluster was proved to be between 24 h and 36 h (Fig. 2e). Conclusively, VRS-bAHL showed reliability in characterizing complex gene expression and regulation with high sensitivity in both solid and liquid cultivated strains.

**VRS-bAHL-mediated characterization of *ang* gene cluster activation.** In addition to oviedomycin, our previous study revealed that tylosin analog compounds were also produced in ΔwblA, a pleiotropic regulatory gene (*wblA*) disruption mutant of *S. ansochromogenes* 7100[38]. The corresponding BGC was identified (Fig. 3a), and renamed as *ang* in this study for its high identity with angolamycin BGC[39,40]. To demonstrate if VRS-bAHL can facilitate the characterization of the activated secondary metabolite BGCs, an *ang* expression reporter strain

7100ang-cviI was constructed, in which P*ang1*, the promoter of a key structural gene *ang1* (also named as *ctg1_1275*), was used to drive *cviI*. 7100ang-cviI was cultivated on eight types of agar media followed by AHL detection using double-layer plate method (Fig. 3b). The results indicated that purple pigment of CV026 was generated around 7100ang-cviI strain grown on TSB agar medium only (Fig. 3c), suggesting that *ang* gene cluster might be activated. The results proved the feasibility of VRS-bAHL in quick screening of activation conditions for a specific gene cluster, avoiding troublesome operations, such as RT-PCR analysis of genes or isolation and identification of products.

To verify the findings above, similar experiments were repeated with *S. ansochromogenes* 7100 wild-type strain, and the fermentation products on some of the representative agar media were analyzed by HPLC. Compound 2, one of the reported *ang*

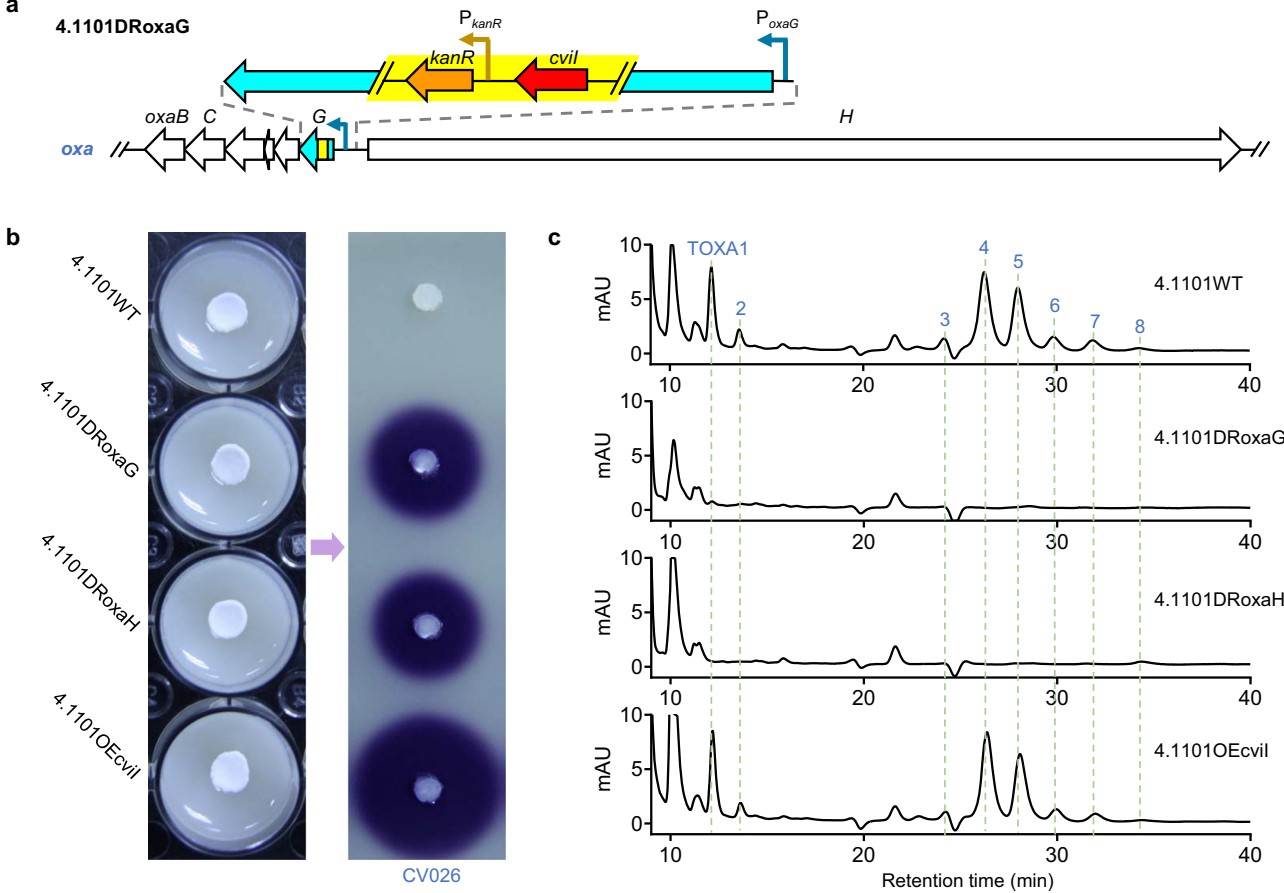

**Fig. 4 Construction and evaluation of dual-functional reporter strains for *oxa*. a** Construction of the dual-functional reporter strains. Taking 4.1101DRoxaG as an example, *cviI* (reporter gene) and *kanR* (kanamycin resistance gene, the selection marker) were ligated as a reporter gene cassette to replace part of the key coding region of *oxaG* via homologous recombination to generate dual-functional reporter strain 4.1101DRoxaG. Another reporter strain 4.1101DRoxaH was constructed similarly. **b** *cviI* expression analyses of 4.1101DRoxaG and 4.1101DRoxaH grown on MS agar medium in 12-well plates by agar plug assays. 4.1101WT (the wild-type strain of *S. longshengensis* CGMCC 4.1101) and 4.1101OEcviI (a derivative strain of 4.1101WT containing *cviI* overexpression plasmid driven by P$_{hrdB}$ on pIJ10500K) were used as negative and positive control, respectively. **c** The production of TOXAs in the above four strains grown on MS agar medium. The differential chromatographic peaks of 4.1101WT and 4.1101OEcviI compared with the reporter strains are indicated by dashed lines, and the corresponding compounds were named as TOXA1-TOXA8, respectively. Data shown in **b** and **c** are representative of two independent experiments that displayed similar results.

products[38], was displayed on HPLC of *S. ansochromogenes* 7100 grown on TSB medium only (Fig. 3d), and its molecular ion ([M + Na]$^+$ *m/z* 607.3463) was further confirmed by liquid chromatography-high resolution electrospray ionization-mass spectrometry (LC-HR-MS) analysis (Supplementary Fig. 4). It was convinced that VRS-bAHL is an efficient approach for screening the activated secondary metabolite BGCs.

**VRS-bAHL-guided genetic identification and yield improvement of oxazolomycin.** Genome sequencing has revealed that almost all *Streptomyces* genomes contain dozens of secondary metabolite BGCs, providing valuable natural product resources. In this study, an oxazolomycin BGC *oxa* (accession number OM807121 in GenBank database) was identified in the genome of *S. longshengensis* CGMCC 4.1101 through antiSMASH analysis[41]. Sequence alignment further indicated that *oxa* showed some similarity with the known oxazolomycin BGC *ozm* in *S. albus* JA3453[42] (Supplementary Fig. 5 and Supplementary Data 1).

To identify the products of *oxa* gene cluster, we constructed two dual-functional reporter strains, 4.1101DRoxaG and 4.1101DRoxaH, in which the key structural genes, *oxaG* and *oxaH* in *oxa*, were disrupted by replacing their partial sequence with *cviI*

reporter gene (Fig. 4a). The generated reporter strains were not only used to determine the transcriptional level of *oxa* via *cviI* expression, but also served as *oxa* deficient mutants for distinguishing the product peaks relating to the gene cluster by comparing the HPLC profile with that of the wild-type (WT) strain. Both reporter strains were cultivated on MS agar medium in 12-well plates and the AHL produced by the reporter strains was determined using agar plug assay. Compared with *S. longshengensis* CGMCC 4.1101 wild-type strain (4.1101WT), both reporter strains (4.1101DRoxaG and 4.1101DRoxaH) induced violacein production in CV026 as the positive control strain (4.1101OEcviI, a derivative strain of 4.1101WT with *cviI* overexpression driven by P$_{hrdB}$ on pIJ10500K) (Fig. 4b). These results indicated that *oxa* was expressed under the above culture conditions. To verify what kind of products might be synthesized by *oxa*, HPLC analyses of the culture extracts from 4.1101WT, 4.1101OEcviI, 4.1101DRoxaG and 4.1101DRoxaH were performed. It was revealed that eight chromatographic peaks of 4.1101WT and 4.1101OEcviI disappeared on the chromatograms of 4.1101DRoxaG and 4.1101DRoxaH, suggesting that the corresponding compounds (named as TOXA1-8, respectively) may be related to *oxa*, but their production was found extremely low (Fig. 4c).

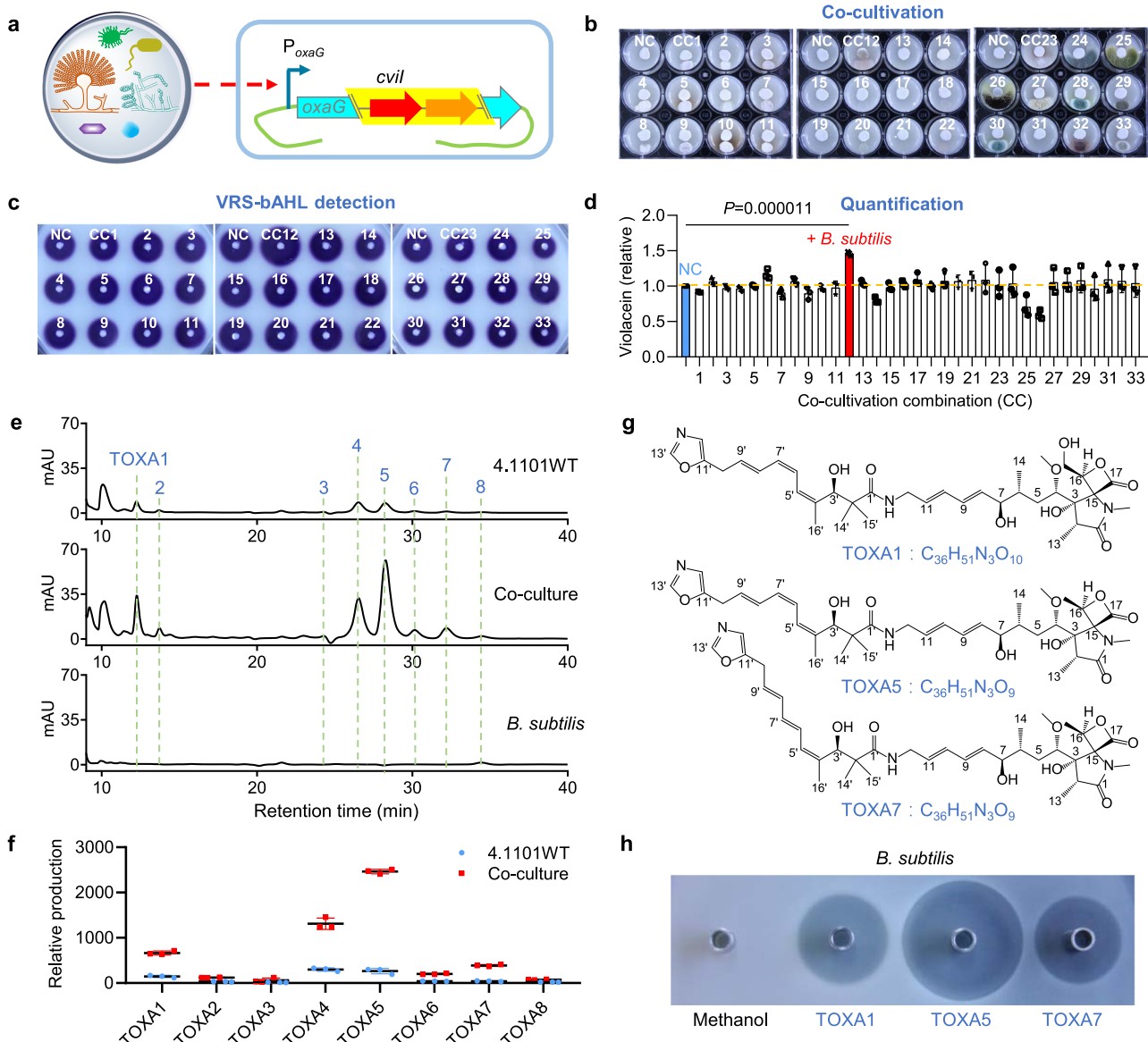

**Fig. 5 VRS-bAHL-guided improvement of oxazolomycin production in *S. longshengensis*. a** Schematic illustration of VRS-bAHL-guided overexpression of *oxa* gene cluster via microbial co-cultivation using dual-functional reporter strain, 4.1101DRoxaG. **b** Co-cultivation of 4.1101DRoxaG with different microorganisms on MS agar medium in 12-well plates. NC, 4.1101DRoxaG was cultivated alone as control. Co-cultivation was conducted by inoculating 4.1101DRoxaG with each specified microorganism adjacent to each other. **c** Detection of AHL production in NC and each co-cultivation combination (CC) through agar plug assays. **d** Quantification of violacein production by the relative area of violacein zone with ImageJ software[59]. All values of CC were normalized to that of NC in each plate. The co-cultivation of 4.1101DRoxaG with *B. subtilis* showed the maximum improvement of AHL production. Individual data points are provided and the data are presented as means ± s.d. (standard deviations) from three biologically independent samples. *P* value was calculated with two-tailed unpaired *t*-test. **e** HPLC analyses of TOXAs production in 4.1101WT, *B. subtilis* and their co-cultivation combination, respectively. **f** Production comparison of TOXAs in 4.1101WT with that in 4.1101WT and *B. subtilis* co-cultivation combination. Individual data points are provided and the data are presented as means ± s.d. from three biologically independent samples. **g** Chemical structures of TOXA1, TOXA5 and TOXA7. **h** Antibacterial assays of TOXA1, TOXA5 and TOXA7 against *B. subtilis*. 40 µg of each compound dissolved in 50 µL of methanol was used for antibacterial assays and 50 µL of methanol was used as blank control. For **b** and **c**, representative images of three biologically independent samples were displayed. For **d** and **f**, source data are provided in Supplementary Data 6. All data shown above are representative of two independent experiments that displayed similar results.

Since the application of various reporter systems greatly facilitated antibiotic titer improvement[9,43–45], and quantification of gene expression by VRS-bAHL is achievable, a high throughput screening method coupling VRS-bAHL with microbial co-cultivation was established to improve TOXAs production (Fig. 5a). 4.1101DRoxaG was co-cultivated with other 33 microorganisms (Supplementary Data 2) on MS agar medium in 12-well plates (Fig. 5b). Violacein production activated by the co-cultivation combinations (CCs) was quantified by comparing the area size of each violacein zone with that of 4.1101DRoxaG cultured alone. The results showed that co-cultivation of 4.1101DRoxaG with *Bacillus subtilis* enhanced violacein production of CV026 obviously (Fig. 5c, d). Similar results were obtained with another dual-functional reporter strain 4.1101DRoxaH (Supplementary Fig. 6). It was implied that the transcription of these genes or the related operons in *S. longshengensis* were

enhanced in the co-culturing system with *B. subtilis*. To further verify the role of *B. subtilis* on promoting *oxa* gene cluster expression, TOXAs production was analyzed by HPLC. The yields of TOXA1-8 in 4.1101WT co-cultured with *B. subtilis* were dramatically increased to 3.3-9.5 folds in comparison with that in 4.1101WT cultured alone (Fig. 5e, f). Subsequently, 7.2 mg of TOXA1, 20 mg of TOXA5 and 3.9 mg of TOXA7 were obtained from 2.5 L of co-cultivation MS agar medium for structural characterizations. LC-HR-MS revealed that the $[M + Na]^+$ ions are *m/z* 708.3482, 692.3519 and 692.3514 for TOXA1, TOXA5 and TOXA7 (Supplementary Fig. 7), and the corresponding molecular formulas were determined as $C_{36}H_{51}N_3O_{10}$, $C_{36}H_{51}N_3O_9$ and $C_{36}H_{51}N_3O_9$, respectively. Based on nuclear magnetic resonance (NMR) analyses (Supplementary Figs. 8–10), TOXA1 was further verified to be a novel oxazolomycin to our knowledge, while TOXA5 and TOXA7 were found to be the reported KSM-2690 B and KSM-2690 C[46], respectively (Fig. 5g). The NMR data were summarized in Supplementary Data 3–5. In addition, antibacterial assays of TOXA1, TOXA5 and TOXA7 were conducted and they exhibited inhibitory activities against various pathogenic bacterial strains, including *B. subtilis*, *Staphylococcus aureus*, *Pseudomonas aeruginosa* and so on (Fig. 5h and Supplementary Fig. 11). So VRS-bAHL is a straightforward and efficient tactic for both characterization and quantification of gene cluster expression at the transcriptional level, facilitating the screening of high-yield antibiotics producers or the optimization of culture conditions via direct comparison of the AHL production.

**Optimization of VRS-bAHL sensitivity**. Visualization and high sensitivity of gene reporter system are particularly favorable for large scale screening, and as a proof of concept study, we applied VRS-bAHL in a pool testing experiment. It was shown that 0.01% of positive strain (a *cviI* overexpression strain in *Streptomyces ansochromogenes*) in a mixture with negative control strain (*Streptomyces ansochromogenes* without *cviI* overexpression) could induce violacein production in CV026, suggesting the great potential of VRS-bAHL for positive strain screening from numerous colonies (Supplementary Fig. 12). However, since VRS-bAHL is hypersensitive, the purple pigment can diffuse extensively, resulting in the overlapping of violacein zones across colonies. To accommodate VRS-bAHL to colony screening and also meet the sensitivity requirement for different purposes, VRS-bAHLs with various sensitivities were constructed based on the affinity of AHL receptors to different AHLs (Fig. 6a).

In brief, *cviI* and *cviI-12472*, AHL synthase genes from *C. violaceum* CV31532 and *C. violaceum* CV12472, respectively, were used as reporter genes[35]. Then we constructed another AHL indicator strain (CV609) through stepwise in-frame deletion of *vioS* and *cviI* of CV31532 (Supplementary Fig. 13a). Unlike CV026, CV609 showed a much lower sensitivity to C6-HSL (Supplementary Fig. 13b). Although the underlying mechanism is not clear yet, CV609 can serve as an indicator strain with reduced sensitivity for short-chain AHLs. Besides, ΔcviI-12472 was also used as an AHL indicator strain[35]. Subsequently, a series of VRS-bAHLs were generated by combining the reporter genes *cviI* and *cviI-12472* with three indicator strains (CV026, ΔcviI-12472 and CV609). Both reporter genes were found to function in *S. ansochromogenes* 7100 according to the production of violacein in CV026. Meanwhile, the zone size and intensity of the purple pigment varied greatly for different combinations of the reporter genes and indicator strains (Fig. 6b).

Then, the efficiency of VRS-bAHLs was further evaluated using a colony screening model. In this experiment, the spores of AHL-producing *Streptomyces* strain (positive control) were mixed with

those of AHL-non-producing strain (negative control), and cultured on MS agar medium followed by double-layer plate method to detect AHL. The positive colonies were readily found out through determining the center of the violacein zones (Fig. 6c). Also, reduced sensitivity (smaller purple zone) was observed in the combination of *cviI-12472* and ΔcviI-12472, which would be helpful for reducing the pigment overlapping. The results confirmed that VRS-bAHL is flexible in sensitivity, which would be beneficial for expanding its utility.

**Discussion**

Bacteria of the genus *Streptomyces* are particularly abundant sources of antibiotics and natural bioactive compounds, providing more than 50% of medically important antimicrobial and antitumor agents[1,47]. Understanding the regulatory mechanisms is very important for activation and expression of the enormous silent secondary metabolite BGCs in *Streptomyces*[48]. So, molecular biological tools, such as reporter systems, play crucial roles in monitoring the transcriptional regulation of genes or gene clusters[6]. However, certain flaws exist in the currently available reporter systems of *Streptomyces*, such as low sensitivity, background interference, potential influences on hosts, requirement of specific substrates and so on[7]. Here, to the best of our knowledge, a new visualization reporter system based on Gram-negative bacterial AHL QS (VRS-bAHL) was established. VRS-bAHL proved efficient for visible and precise evaluation of gene expression in *Streptomyces*.

The most prominent feature of VRS-bAHL is high sensitivity and low background interference. Its hypersensitivity is attributed to the sensitive response of the receptors to AHLs in *C. violaceum*, and nanomolar concentrations of AHL would be able to activate violacein production or bioluminescence generation of indicator strains[35]. More strikingly, there was no obvious background interference observed for VRS-bAHL. One of the potential interference could be from γ-butyrolactone signaling molecules of *Streptomyces* since they share some structural similarity with AHLs, and their influence on QS of *C. violaceum* was indicated previously. However, since the effective concentration of γ-butyrolactones inducing violacein generation is far beyond that of AHL[35], they would have no or negligible interference on the performance of VRS-bAHL. In addition, the potential inhibitory activity of bioactive metabolites from *Streptomyces* against *C. violaceum* might be present, but it could be attenuated via dilution or reducing the loading amount of samples onto indicator strains.

VRS-bAHL is also advantageous in many other aspects. The reporter gene, AHL synthase gene *cviI* or *cviI-12472* is just 654 bp in size, allowing its efficient expression in most microbial host cells. Furthermore, the biosynthesis of AHLs generally requires only two precursors, SAM and acyl-ACP[33], both of which are ubiquitous in microorganisms, so supply of other specific substrates is not required. Hence, as evidenced in the present study, VRS-bAHL is eventually suitable for most *Streptomyces* strains grown on different media. More interestingly, the strength of violacein pigment was positively correlated with the antibiotic production in *Streptomyces*, which is essential for quantification. Finally, the compatible coupling of VRS-bAHL with various mutagenesis or gene engineering strategies, such as UV mutagenesis, ribosome engineering, co-cultivation and so on, can be envisioned for screening high-yield antibiotics producers or their elicitor molecules[45,49–51].

Since visible reporter systems are particularly favorable in large-scale characterization and evaluation of gene expression, VRS-bAHL was accommodated into colony screening on agar medium. The hybrid VRS-bAHLs constructed via combining

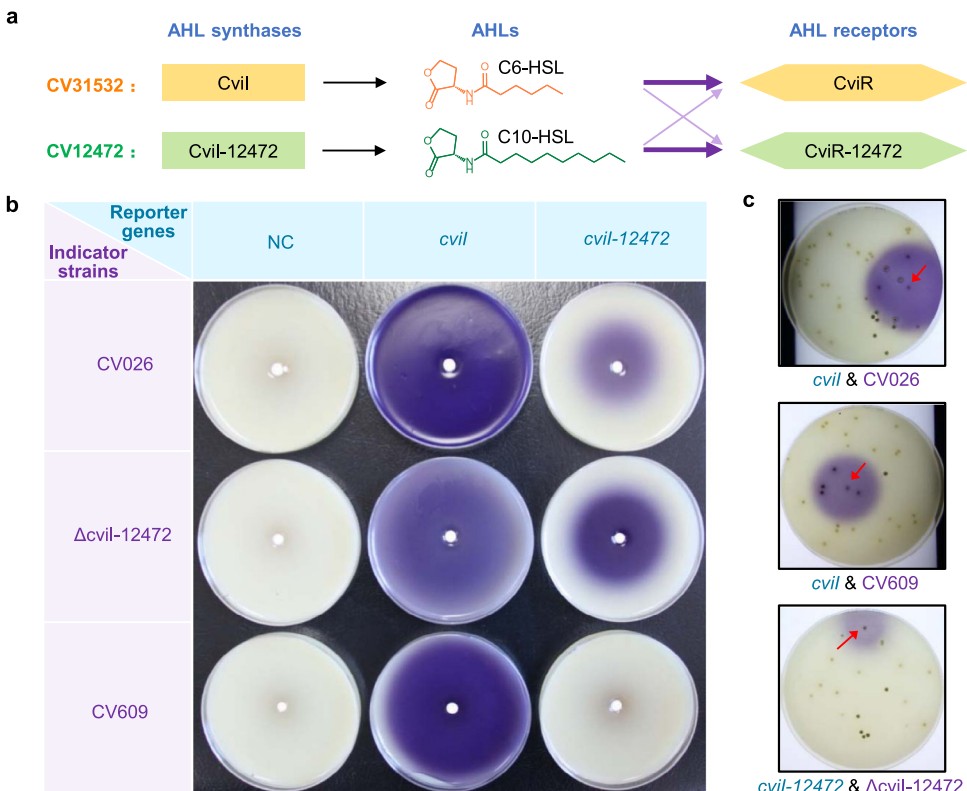

**Fig. 6 Construction and validation of VRS-bAHLs with various sensitivities. a** Schematic illustration of VRS-bAHL with different sensitivities. The construction of VRS-bAHL with different sensitivities was based on the cross-talk between the AHL QS signaling systems of CV31532 and another *C. violaceum*, CV12472, and the latter mainly employs long-chain AHL (C10-HSL) to induce violacein production. CV31532 and CV12472 can respond to AHLs from each other, but with lower sensitivities[60]. As a result, combinations of AHL synthase gene of CV31532 or CV12472 with different indicator strains (the AHL and violacein deficient mutants of CV31532 or CV12472) can generate a series of VRS-bAHLs with changeable sensitivities. **b** Detection and comparison of VRS-bAHL sensitivity. Two reporter genes, *cviI* and *cviI-12472*, were constitutively expressed under the control of P$_{hrdB}$ on plasmid pIJ10500K in *S. ansochromogenes* 7100 and the AHL production was detected by double-layer plate method with three indicator strains (CV026, ΔcviI-12472 and CV609), respectively. The derivative strains of *S. ansochromogenes* 7100 used here are as follows: AHL-producing strains, 7100OEcviI and 7100OEcviI-12472 (derivative strains of *S. ansochromogenes* 7100 containing *cviI* or *cviI-12472* overexpression plasmid driven by P$_{hrdB}$ on pIJ10500K), and AHL-non-producing strain 7100NC (*S. ansochromogenes* 7100 containing blank pIJ10500K). **c** A proof of concept experiment to investigate the application of VRS-bAHLs in colony screening. In brief, the spores of 7100OEcviI or 7100OEcviI-12472 were mixed with those of 7100NC at a ratio of 1/100 and the diluted mixtures were inoculated on MS agar medium. After cultivation for three days, double-layer plate method was applied to detect the colonies inducing violacein production, and the sensitivity of different VRS-bAHLs was compared. The VRS-bAHLs used for these tests are as follows: *cviI* & CV026, *cviI* as the reporter gene and CV026 as the AHL indicator strain; *cviI* & CV609, *cviI* as the reporter gene and CV609 as the AHL indicator strain; *cviI-12472* & ΔcviI-12472, *cviI-12472* as the reporter gene and ΔcviI-12472 as the AHL indicator strain. The positive control in the mixture was indicated with red arrows. Data shown in **b** and **c** are representative of two independent experiments that displayed similar results.

reporter genes with different indicator strains exhibited changeable sensitivity, which led to minimal overlapping of the pigment zones across colonies. The target clones identified through determining the center of pigment zones can be recovered from the plate via two approaches. The first option is to directly pick them out and inoculate onto a fresh plate containing nalidixic acid to remove indicator strain *C. violaceum* but allow *Streptomyces* to grow. Alternatively, plate replication may be conducted prior to covering it with indicator strains, and the target colonies can be readily picked out on the replicate plates[6]. Thus, different sensitivities of VRS-bAHLs would overcome the potential inconvenience resulted from the hypersensitivity for colony identification from numerous communities, conferring this strategy even broader applicability.

To our knowledge, this is the first time that a robust visualization reporter system was ingeniously constructed based on Gram-negative bacterial AHL QS. This system possesses high sensitivity, broad compatibility and clean background, the unique and valuable characteristics particularly beneficial for characterizing the expression and activation of silent gene clusters,

therefore promoting the discovery of natural products in *Streptomyces* and eventually many other microorganisms.

## Methods

**Bacterial strains, plasmids, primers and general growth conditions**. Bacterial strains, plasmids and primers used in this study are listed in Supplementary Data 7–9, respectively.

To prepare spores, *S. virginiae*, *S. venezuelae* and their derivative strains were grown on MYM agar medium[52], other *Streptomyces* were grown on MS agar medium[6], and all fungi were grown on PDA agar medium[53]. For growth and fermentation, *Streptomyces* were usually cultured on MM (mannitol as carbon source), GP, ISP2, SMMS, MYM, MS, YEME and TSB (BD Bacto TM) agar media or in liquid MS and SP meida[6,36,54]. Unless otherwise stated, all cultivations were carried out at 28 °C. 15 g L$^{-1}$ and 10 g L$^{-1}$ agar were used in the normal and soft agar medium, respectively. Antibiotics used for selection and plasmid maintenance were as follows: 10 μg mL$^{-1}$ apramycin for *S. ansochromogenes*, 60 μg mL$^{-1}$ apramycin for *S. coelicolor*, 75 μg mL$^{-1}$ apramycin for *S. longshengensis* and 50 μg mL$^{-1}$ apramycin for other *Streptomyces*; 10 μg mL$^{-1}$ kanamycin for *S. ansochromogenes* and 50 μg mL$^{-1}$ kanamycin for other *Streptomyces*; 25 μg mL$^{-1}$ nalidixic acid for all *Streptomyces*; 100 μg mL$^{-1}$ hygromycin, 100 μg mL$^{-1}$ kanamycin, 25 μg mL$^{-1}$ chloramphenicol and 100 μg mL$^{-1}$ apramycin for *E. coli*; 50 μg mL$^{-1}$ kanamycin for *C. violaceum*.

**Construction of plasmids and strains**. The plasmids used in *Streptomyces* were constructed in *E. coli* JM109 and then conjugally transferred into corresponding *Streptomyces* via *E. coli* ET12567/pUZ8002. The plasmids used in *C. violaceum* were constructed in *E. coli* S17-1 λpir, which were then used for conjugal transfer from *E. coli* into *C. violaceum*[35].

pSET152 and its derivative plasmid pSPhrdB-cviI were used in the feasibility evaluation of VRS-bAHL in seven different *Streptomyces*. In all other experiments, unless otherwise stated, pIJ10500K and its derivative plasmids were used for gene expression.

Construction of pSPhrdB-cviI and the corresponding *Streptomyces* derivatives: the promoter P$_{hrdB}$ was amplified by PCR with primer pair SPHRDBF/PHRDBR using the genomic DNA of *S. coelicolor* as template, and then digested with *Xba*I. Subsequently, reporter gene *cviI* was amplified by PCR with primer pair I532F(phosphorylated)/SI532R using the genomic DNA of CV31532 as template and digested with *Bam*HI. Finally, the promoter and reporter gene obtained above and the digested pSET152 DNA fragment with *Xba*I/*Bam*HI were ligated together to generate the corresponding plasmid. They were then introduced into *Streptomyces* and integrated at the *attB* site on the genome to generate corresponding NC or OEcviI strains.

Construction of pIJ10500K-derived plasmids and the corresponding *Streptomyces* derivatives: first, pIJ10500K was constructed based on pIJ10500[55]. The fragment containing *kanR*, a kanamycin resistance gene, was amplified by PCR with primer pair KANRF/KANRR and using the plasmid pCS26-Pac[56] as template followed by digestion with *Hin*dIII/*Kpn*I. The resulting fragment was inserted into *Hin*dIII/*Kpn*I sites of pIJ10500 to generate pIJ10500K. For construction of pPhrdB-cviI, pPhrdB-cviI-12472, pPovmOI-cviI and pPang1-cviI, the promoters (P$_{hrdB}$, P$_{ovmOI}$ or P$_{ang1}$) were amplified by PCR with primer pairs PHRDBF/PHRDBR, POVMF/POVMR or PANGF/PANGR using the genomic DNA of *S. coelicolor* or *S. ansochromogenes* as templates and then digested with *Nde*I. Subsequently, reporter genes, *cviI* or *cviI-12472*, were amplified by PCR with primer pairs I532F(phosphorylated)/I532R or I472F(phosphorylated)/I472R using the genomic DNA of CV31532 or CV12472 as template, respectively, and digested with *Spe*I. Finally, the promoters and reporter genes obtained above and the digested pIJ10500K DNA fragment with *Nde*I/*Spe*I were ligated to generate the corresponding plasmids. They were then introduced into *Streptomyces* and integrated at the *attB* site on the genome to generate *Streptomyces* derivatives.

Construction of *Streptomyces* reporter strains containing pKC1139 or pKC1139::P$_{hrdB}$ZW: the two plasmids were introduced into M1146ovm-cviI or 7100ovm-cviI through conjugal transfer to generate M1146ovm-cviI-NC, M1146ovm-cviI-OEzw, 7100ovm-cviI-NC or 7100ovm-cviI-OEzw, respectively.

Construction of dual-functional reporter strains of *oxa*: for construction of plasmid pKDRoxaG, the upstream and downstream regions of *oxaG* were obtained by PCR amplification with the primer pair DROXAGLF/DROXAGLR or DROXAGRF/DROXAGRR using the genomic DNA of *S. longshengensis* as template. The *cviI* fragment containing its own RBS DNA sequence was amplified by PCR with primer pair RBSI532F/RBSI532R using the genomic DNA of CV31532 as template. The *kanR* fragment containing its own promoter region was amplified by PCR using primer pair PKANRF/PKANRR from pCS26-Pac. Finally, the above four fragments were ligated with *Eco*RV-digested pKC1139 via Gibson Assembly to generate pKDRoxaG[57]. pKDRoxaH was constructed similarly using the corresponding primers. The two plasmids were then introduced into *S. longshengensis*, respectively, and double-crossover mutants were screened to obtain the dual-functional reporter strains (4.1101DRoxaG and 4.1101DRoxaH).

Construction of 7100ovm-cviIΔsabA: *sabA* was disrupted in 7100ovm-cviI via homologous recombination using plasmid pKC1139AD as previously described[58].

In order to obtain another AHL indicator strain CV609, the plasmid pKKDvioS for disruption of *vioS* was constructed first. The upstream and downstream regions of *vioS* were obtained by PCR amplification with the primer pair DVIOSLF/DVIOSLR or DVIOSRF/DVIOSRR using the genomic DNA of CV31532 as template. The above two fragments were ligated with *Eco*RV-digested pKnock-Km via Gibson Assembly to generate pKKDvioS. pKKDvioS was then introduced into CV31532 followed by screening of the double-crossover mutants to obtain ΔvioS. Subsequently, *cviI* was disrupted in ΔvioS with plasmid pKKDcviI in the similar way to generate the AHL indicator strain named as CV609 (ΔvioS/ΔcviI).

**Detection of VRS-bAHL**. Double-layer plate method: firstly, *Streptomyces* derivatives were inoculated on the surface of agar media and cultivated for appropriate time (about 2–3 days). Then, CV026 in LB soft agar medium (at a ratio of 10%) was overlaid on the top of the agar medium plates of *Streptomyces* strains and further cultivated for appropriate time (about 20 h). The expression of reporter genes was evaluated according to violacein production in CV026.

Agar plug or Oxford cup assay is also based on violacein production of CV026 as used in double-layer plate method. LB soft agar medium plates containing CV026 (at a ratio of 10%) were prepared. The agar plugs with *Streptomyces* or Oxford cups containing liquid samples were placed on the top of CV026 grown in soft agar medium plates. After incubation for appropriate time, the expression of reporter genes was evaluated according to violacein production in CV026.

AHL detection by co-cultivation: both *Streptomyces* derivatives and CV026 were inoculated on the surface of agar medium plates at adjacent positions. The expression of reporter genes was evaluated according to violacein production in CV026 nearby. This is more suitable for real-time monitoring of gene expression in *Streptomyces* during growth.

Quick AHL detection with *E. coli* reporter strain (EJ532-4): In EJ532-4, AHL receptor gene *cviR* was constitutively expressed and the luciferase coding genes were under the control of P$_{vioA}$, a CviR target promoter. The expression of luciferase coding genes could be activated in the presence of AHL, leading to the generation of bioluminescence[35]. In this study, for detection of AHL in *Streptomyces* cultures, the samples were added into liquid LB medium containing EJ532-4 at a ratio of 10% in 96-well plates followed by about 1 h of incubation. The intensity of bioluminescence could be measured in endpoint or continuous reading mode with plate readers to determine the expression level of reporter genes.

**Preparation and analysis of compounds**. For tylosin analogs analyses, *S. ansochromogenes* 7100 was cultivated on agar media for 7 days and extracted with equal volume of methanol. The extracts were then concentrated and re-dissolved in methanol for HPLC analysis as previously reported[38]. For oviedomycin analyses, 7100ovm-cviI or 7100ovm-cviIΔsabA was fermented in liquid SP media containing mannitol or glucose as carbon source for appropriate time and the fermentation broths were filtered for oviedomycin analysis by HPLC as previously described[37]. For TOXAs analyses, *S. longshengensis* wild-type strain or its derivatives were cultivated alone or with other microorganisms together on MS agar medium in 12-well plates for three days followed by extraction with equal volume of methanol. The extracts were analyzed by HPLC using Zorbax SB-C18 column (4.6 × 250 mm, 5 μm) with a flow rate of 1 mL min$^{-1}$ at 280 nm detection wavelength. The elution was performed with a concentration gradient change of acetonitrile (ACN) in de-ionized water containing 0.1% formic acid and the detailed elution profiles of HPLC are available in Supplementary Table 1.

For TOXAs preparation and identification: *S. longshengensis* wild-type strain was co-cultivated with *B. subtilis* on MS agar medium in normal 90 mm diameter petri dishes for three days, and then the solid culture was chopped and extracted with equal volume of methanol. The extract was concentrated and re-dissolved in 50% methanol in water. TOXAs were isolated from the extract through two rounds of HPLC fractionations using Zorbax SB-C18 column (9.4 × 250 mm, 5 μm) with a flow rate of 3 mL min$^{-1}$ at 280 nm detection wavelength. The same HPLC conditions were applied as mentioned above except that de-ionized water without formic acid was used in the second round of HPLC collection. Comparisons of the HPLC with that of the dual-functional reporter strain 4.1101DRoxaG (no oxazolomycin can be produced due to *oxaG* disruption) were performed, and the differential peaks corresponding to oxazolomycins on HPLC of WT strain were collected and concentrated for structure determination and bioactivity assays. Mass spectral analyses were performed on AGILENT 1200HPLC/6520QTOFMS in positive mode. NMR spectra were recorded on a 500-MHz Bruker spectrometer using CD$_3$SOCD$_3$ as solvent.

**Bioassays of TOXAs**. Bioassays of TOXAs were carried out as previously reported and the bacterial strains were cultivated in LB soft agar medium[38].

**Statistics and reproducibility**. All images shown in this paper are representative of similar conclusions seen in at least two independent experiments. All graphing was done with GraphPad Prism 8.0.1 unless otherwise stated and means of three biological independent samples are shown. Error bars represent standard deviations. All *P* values were calculated with two-tailed unpaired *t*-test. All data are representative of at least two independent experiments. Randomization is not relevant to this study. Sample size estimation was based on widely used ones of previous publications to ensure that they are sufficient enough to achieve significant results. No data were excluded from the analyses and all attempts at replication were successful.

**Reporting summary**. Further information on research design is available in the Nature Research Reporting Summary linked to this article.

## Data availability

All data supporting this work are available within this article and the Supplementary Information. The sequence data of *oxa* gene cluster have been deposited in NCBI GenBank with the accession number OM807121, and sequences data about *ovm* and *ang* gene clusters can be found from the genomic DNA of *S. ansochromogenes* 7100 which is available in China National Microbiology Data Center (NMDC) with accession number of NMDC60029072. Sequences of all newly generated plasmids have been deposited in NMDC with accession numbers of NMDCN00011OL–NMDCN00011OU. Source data of Fig. 5d, f and Supplementary Figs. 2, 3b and 6b are provided in Supplementary Data 6 and 10–12. Further information or materials related to the findings of this study are available from the corresponding authors upon reasonable request.

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

## Acknowledgements

This work was supported by grants from the National Key Research and Development Program of China [grant numbers 2020YFA0907800 and 2018YFA0901900]; the National Natural Science Foundation of China [grant number 82173720]; and Beijing Natural Science Foundation [grant number 7212153]. We thank Drs. Guomin Ai, Wenzhao Wang and Jinwei Ren (Institute of Microbiology, Chinese Academy of Sciences, Beijing, China) for assistance with mass spectrometry or NMR analyses. We thank Professor Dongjie Tang (College of Life Science and Technology, Guangxi University, China) for kindly offering *Xanthomonas campestris* Xcc 8004.

## Author contributions

X.L. performed most of the experiments and wrote the original manuscript draft. Jine Li performed data analysis and revised the manuscript. Y.L. performed partial experiments of plasmid and strain constructions. Junyue Li, H.S. and J. Zheng performed partial experiments of VRS-bAHL optimization. J. Zhang supervised the research work and revised the manuscript. H.T. conceived and supervised the whole research work and also revised the manuscript. All authors have read and agreed to publish the manuscript.

## Competing interests

The authors have filed two patents for this work to the China National Intellectual Property Administration. Huarong Tan, Xiang Liu, Jihui Zhang, Yue Li and Jine Li are inventors of the first patent application (Chinese patent application number 202111355337.0). Huarong Tan, Xiang Liu, Jihui Zhang, Huiying Sun and Jine Li are inventors of the second patent application (Chinese patent application number 202111356220.4). All other authors declare no competing interests.
