## [Peer Review File · Communications Biology]

Reviewers' comments:

Reviewer #1 (Remarks to the Author):

This article deals with the quorum sensing based reporter system utilizing violacein production. Findings are important for its application as genetic tool. But there are several issues which needs to be addressed before being considered for publishing.

1. Violacein also exhibits various biological functions such as antimicrobial property for gram positive bacteria. This might influent its reporter efficiency from gram positive bacteria.
2. In the text the author mentioned that the current systems (such as GFP or xylE) are often dramatically interfered by the secondary metabolites from *Streptomyces*. However, quorum sensing also presented complex regulation and have high correlation of secondary metabolites. Please give the more detail information of this in the introduction.
3. The sensitivity of this system should be evaluated. For example, utilizing AHL addition to induce the violacein production in this system.
4. The quorum sensing (QS) is important for microbe communication. I think the system provide great innovation for QS detection. However, the author tried to connect antibiotic production and QS induction. This may cause misunderstanding for reader. Please provide more detail information in the introduction.
5. The document can be improved if revised by an English editor.

Reviewer #2 (Remarks to the Author):

Manuscript by Liu at al. reports the development and testing of a new reporter system based on the quorum sensing of the Gram-negative bacterium *Chromobacterium violaceum*. This system employs a reporter gene *cviI* essential for the production of an acyl-homoserine lactone (AHL), which is transcriptionally coupled to secondary metabolite biosynthesis genes in *Streptomyces* bacteria. Hence, when the latter are expressed, the co-expression of *cviI* leads to production of AHL which, upon diffusion into the overlaid culture of *C. violaceum* triggers production of pigmented compound violacein.

The overall idea in itself is original, but the usefulness of this reporter system, especially for selection of secondary metabolite overproducers is questionable. Here are the major concerns:

- 1) Activation of biosynthetic gene transcription does not necessarily mean secondary metabolite production. Also, absence of expression of even one gene not coupled to *cviI* may block the whole pathway. Regulation of translation, post-translational modifications and precursor supply may play decisive roles. Hence, biosensor system based on detection of transcription is prone to flaws and could not be as efficient as the authors claim.
- 2) Physiology of bacteria when grown on solid and in liquid media are very different, but the developed system is based on agar media cultivation only. Authors must at least demonstrate the system utility after selected strains are cultivated in liquid medium.
- 3) In their proof-of-principle experiments with oxazolomycin cluster authors disrupt the oxa biosynthetic genes, rendering the strains non-producers. How relevant is this for the selection of strains with better production? Also, very few details are provided on how the authors managed to extract enough oxazolomycin and its congeners from agar-grown cultures for NMR.

Reviewer #3 (Remarks to the Author):

In the article entitled "A visualization reporter system with high-sensitivity for characterizing expression of antibiotics biosynthetic gene clusters," the authors present a new visualization reporter system. The tool described is intended for the characterization of promoters in Gram-positive bacteria,

especially for the identification of new secondary metabolites. This system can be used to: (I) determine the conditions under which previously silenced biosynthetic clusters of unknown products are activated, (II) investigate the intricate regulatory systems for the synthesis of secondary metabolites, and (III) optimize the synthesis of these compounds.

The authors present the idea of this system in a clear and lucid way. They present several experiments in which they prove the applicability of this tool in the mentioned above situations. In the final part of the article they present a version of the system optimized for screening assays.

The presented solution is new and not previously published. It is based on an ingenious idea of using acyl-homoserine lactones - molecules present in Quorum Sensing systems in Gram-negative bacteria - for testing in Gram-positive bacteria. Quorum Sensing systems in Gram-positive bacteria are usually based on gamma-butyrolactones. Such a solution makes the presented system insensitive to background interfering factors present in Gram-positive bacteria.

In my opinion, this work will be of interest not only to the community associated with the analysis and search for new antibiotics but also to all microbiologists because the presented idea has a potential beyond the applications described in the article. On the basis of this idea many new applications can be created.

Text-related comments:

1. The discovery of a novel oxazolomycin TOXA1 should be mentioned in the abstract. Confirmation of presented tool with the discovery of a new version of oxazolomycin increases the value of this work. Therefore, it should be presented to a wide audience.

2. The system presented is excellent for qualitative testing. The authors mention method of its application in quantitative measurements based on direct measurement of violacein (figure 1, text line 126). Is it possible to use this test in quantitative measurements? It would be good if the authors comment on this in more detail in the main text, and propose a simple application for quantitative tests, e.g. with a plate reader. This would greatly expand the universality of the proposed tool.

3. It would also be interesting to conduct continuous measurements. Perhaps the authors, on the basis of their experience, have some proposal of a method that would allow for this, and they can sketch it. It would be good to have a method in which one can compare the timing of activation or deactivation of promoters activity in parallel cultures. As it has been shown (e.g. in *S. coelicolor*) activation systems of individual polyketide clusters can be dependent on each other and in the searching and awakening of new polyketide clusters the knowledge of mutual relations is important.

4. The authors should draw the reader's attention to the fact that the mutant they used, M1146, has 4 polyketide clusters removed but leaves the GBL butanolide system and the SARP regulator CpkN intact. Recently, it has been shown that the two elements are related and are important in the global regulatory system. The authors should make sure that there is no interference between the butanolide system and their reporter system. At the very least, this needs to be commented in the text

Reviewers' comments:

Reviewer #1 (Remarks to the Author):

This article deals with the quorum sensing based reporter system utilizing violacein production. Findings are important for its application as genetic tool. But there are several issues which needs to be addressed before being considered for publishing.

Response: We thank Reviewer #1's comments and suggestions.

1. Violacein also exhibits various biological functions such as antimicrobial property for gram positive bacteria. This might influent its reporter efficiency from gram positive bacteria.

Response: It's true that violacein can have antimicrobial activity against Gram-positive bacteria. But this is not an issue for VRS-bAHL reporter system because induction of violacein production in CV026 by AHL produced in *Streptomyces* species was taken place after the *Streptomyces* had grown to the expected stages. Moreover, violacein is mainly accumulated intracellularly, so the effect can be circumvented.

2. In the text the author mentioned that the current systems (such as GFP or xylE) are often dramatically interfered by the secondary metabolites from *Streptomyces*. However, quorum sensing also presented complex regulation and have high correlation of secondary metabolites. Please give the more detail information of this in the introduction.

Response: Quorum sensing (QS) of microorganisms does have complex correlation with microbial secondary metabolites. As suggested, more descriptions were added in the introduction (P4, lines 73-78). However, very interestingly, the influence of AHL on secondary metabolites biosynthesis of *Streptomyces* was not observed in all seven *Streptomyces* species tested. We speculated that this could be due to the absence of AHL receptor (whereby AHL exerts its function) homologous proteins in most *Streptomyces* according to reported studies. What's more, although some secondary metabolites from *Streptomyces* may affect QS of *Chromobacterium violaceum*, their effective concentrations are generally much higher than that of AHL on the QS of indicator strain CV026, and their potential effects could be attenuated via dilution or reducing the loading amount of samples. In addition, the actual performance of

VRS-bAHL in the tested *Streptomyces* strains and different application scenarios proved its feasibility.

3. The sensitivity of this system should be evaluated. For example, utilizing AHL addition to induce the violacein production in this system.

Response: As suggested, the sensitivity of VRS-bAHL was evaluated with AHL addition (please see the newly added Supplementary Fig. 3), in which 10 nM of C6-HSL could trigger violacein production in CV026. The sensitivity evaluation was also shown in Supplementary Fig. 12b. In addition, a pool testing experiment was designed to further evaluate the sensitivity of VRS-bAHL in complex mixture samples. We found that 0.01% of positive strain (a *cviI* overexpression strain in *Streptomyces ansochromogenes*) in a mixture with negative control strain (*Streptomyces ansochromogenes* without *cviI* overexpression) could induce violacein production in CV026 (see the newly added Supplementary Fig. 11), proving the sensitivity and feasibility of VRS-bAHL in positive strain screening from numerous colonies. The detailed descriptions have been added in the manuscript (P6, lines 120-122; P19, lines 350-355).

4. The quorum sensing (QS) is important for microbe communication. I think the system provide great innovation for QS detection. However, the author tried to connect antibiotic production and QS induction. This may cause misunderstanding for reader. Please provide more detail information in the introduction.

Response: QS systems are important for microorganisms and have been widely exploited as genetic tools. In this study, the QS-based reporter system VRS-bAHL was mainly set up to indicate antibiotic biosynthetic gene expression in heterologous hosts, *Streptomyces*, from gene transcription level based on reporter gene *cviI* expression in *Streptomyces* reporter strains. More details about QS and VRS-bAHL have been added in the introduction (P4, lines 73-78 and 82-83).

5. The document can be improved if revised by an English editor.

Response: The manuscript has been further revised and improved.

Reviewer #2 (Remarks to the Author):

Manuscript by Liu et al. reports the development and testing of a new reporter system

based on the quorum sensing of the Gram-negative bacterium *Chromobacterium violaceum*. This system employs a reporter gene *cviI* essential for the production of an acyl-homoserine lactone (AHL), which is transcriptionally coupled to secondary metabolite biosynthesis genes in *Streptomyces* bacteria. Hence, when the latter are expressed, the co-expression of *cviI* leads to production of AHL which, upon diffusion into the overlaid culture of *C. violaceum* triggers production of pigmented compound violacein.

The overall idea in itself is original, but the usefulness of this reporter system, especially for selection of secondary metabolite overproducers is questionable. Here are the major concerns:

Response: We appreciate Reviewer #2's comments and suggestions.

1) Activation of biosynthetic gene transcription does not necessarily mean secondary metabolite production. Also, absence of expression of even one gene not coupled to *cviI* may block the whole pathway. Regulation of translation, post-translational modifications and precursor supply may play decisive roles. Hence, biosensor system based on detection of transcription is prone to flaws and could not be as efficient as the authors claim.

Response: Activation of antibiotic biosynthetic gene clusters (BGC) does require successful transcription and translation of all related genes, and appropriate precursor supplies are also very important. VRS-bAHL serves as a tool for quick screening of the potentially activated BGC at transcriptional level, or screening of culturing conditions. Then other approaches would be applied to further verify the activation of corresponding antibiotic BGC.

In fact, gene expression reporter-guided approaches have been proved successful in screening of the activation of secondary metabolites biosynthesis or their overproducers (Refs. 9, 43, 44, 45, 51, 53). In the present study, one of the examples of VRS-bAHL application on activation of antibiotics was shown in Fig. 4. Besides, we also proved VRS-bAHL feasibility in verifying oviedomycin biosynthesis activation by disruption of butenolide biosynthase gene (*sabA*) in coordination with glucose addition. The newly added results were shown in Fig. 2d and e. For the application of VRS-bAHL in determining high-producing strains or conditions, the evaluation is mainly based on key gene transcriptional level as an initial step, which is eventually to be confirmed by the product analysis. As an example, using dual

reporter strains 4.1101DRoxaG (Fig. 5) and 4.1101DRoxaH (newly added Supplementary Fig. 6), we established the oxazolomycin high-producing system. Some descriptions have been added accordingly (P9, lines 166-178; P16, lines 299-302).

2) Physiology of bacteria when grown on solid and in liquid media are very different, but the developed system is based on agar media cultivation only. Authors must at least demonstrate the system utility after selected strains are cultivated in liquid medium.

Response: VRS-bAHL system can be used in liquid cultivated strain, and the samples obtained can be analyzed using the Oxford cup method as shown in Supplementary Fig. 1c. We also did verification assays with 7100ovm-cvi Δ sabA, another oxiedomcyin producing strain with more complicated regulation and involving carbon source utilization, in which the time course of *ovm* activation was determined (Fig. 2d and e). The related descriptions were added (P9, lines 166-178).

3) In their proof-of-principle experiments with oxazolomycin cluster authors disrupt the *oxa* biosynthetic genes, rendering the strains non-producers. How relevant is this for the selection of strains with better production? Also, very few details are provided on how the authors managed to extract enough oxazolomycin and its congeners from agar-grown cultures for NMR.

Response: The disruption of *oxaG/F* was designed for genetically identifying the oxazolomycin BGC because *oxa* gene cluster and its product in this strain was not identified before. Meanwhile, considering the complexity of HPLC profiles and general difficulties in correlating secondary metabolites peaks on HPLC with their BGC, dual functional strains were constructed. In brief, they contain two basic modules, one is the *cviI*-reporter module, and another one is the disruption of structural gene. With reporter module, we could determine what conditions could activate the expression of BGC; then the HPLC peaks of wild-type (WT) strain and the dual functional strain cultured under this condition could be compared, and the peaks present on the HPLC profile of WT but absent on that of disruption mutant (dual functional strain) would be likely correlated with the BGC. This is a way facilitating the correlation of BGC with products. Indeed, for screening oxazolomycin high-producing conditions, the *cviI* reporter gene module is sufficient for indicating

the gene transcription, while for the production of the antibiotics, structural gene disruption mutant should not be used. We have made revisions in the text to make it clearer (P13, line 243; P14, lines 257-258). In terms of the oxazolomycin isolation, more details were added in the Results and Methods (P16, lines 306-308; P29, lines 602-604 and 610-614).

Reviewer #3 (Remarks to the Author):

In the article entitled "A visualization reporter system with high-sensitivity for characterizing expression of antibiotics biosynthetic gene clusters," the authors present a new visualization reporter system. The tool described is intended for the characterization of promoters in Gram-positive bacteria, especially for the identification of new secondary metabolites. This system can be used to: (I) determine the conditions under which previously silenced biosynthetic clusters of unknown products are activated, (II) investigate the intricate regulatory systems for the synthesis of secondary metabolites, and (III) optimize the synthesis of these compounds.

The authors present the idea of this system in a clear and lucid way. They present several experiments in which they prove the applicability of this tool in the mentioned above situations. In the final part of the article they present a version of the system optimized for screening assays.

The presented solution is new and not previously published. It is based on an ingenious idea of using acyl-homoserine lactones - molecules present in Quorum Sensing systems in Gram-negative bacteria - for testing in Gram-positive bacteria. Quorum Sensing systems in Gram-positive bacteria are usually based on gammabutyrolactones. Such a solution makes the presented system insensitive to background interfering factors present in Gram-positive bacteria.

In my opinion, this work will be of interest not only to the community associated with the analysis and search for new antibiotics but also to all microbiologists because the presented idea has a potential beyond the applications described in the article. On the basis of this idea many new applications can be created.

Response: Many thanks for the comments and suggestions.

Text-related comments:

1. The discovery of a novel oxazolomycin TOXA1 should be mentioned in the abstract. Confirmation of presented tool with the discovery of a new version of oxazolomycin increases the value of this work. Therefore, it should be presented to a wide audience.

Response: As suggested, the discovery of a new oxazolomycin compound has been added in the abstract (P2, lines 26-27).

2. The system presented is excellent for qualitative testing. The authors mention method of its application in quantitative measurements based on direct measurement of violacein (figure 1, text line 126). Is it possible to use this test in quantitative measurements? It would be good if the authors comment on this in more detail in the main text, and propose a simple application for quantitative tests, e.g. with a plate reader. This would greatly expand the universality of the proposed tool.

Response: Yes, it can be used in quantitative measurements via determining the area size of the purple violacein zone using ImageJ software. While using the *E. coli* indicator strain for AHL quantitative measurement in 96-well plates was also feasible (Supplementary Fig. 2). In addition, we performed another quantitative test in 96-well plates to determine the effective concentration of AHL on violacein production, in which the optical density of DMSO extract of violacein was measured with a plate reader. And we can see that violacein production is positively correlated with AHL concentration (Supplementary Fig. 3). More descriptions have been added (P6, lines 120-125).

3. It would also be interesting to conduct continuous measurements. Perhaps the authors, on the basis of their experience, have some proposal of a method that would allow for this, and they can sketch it. It would be good to have a method in which one can compare the timing of activation or deactivation of promoters activity in parallel cultures. As it has been shown (e.g. in *S. coelicolor*) activation systems of individual polyketide clusters can be dependent on each other and in the searching and

awakening of new polyketide clusters the knowledge of mutual relations is important.

Response: The continuous measurements could be achieved preliminarily through co-cultivation assays of *Streptomyces* and indicator strain as shown in Supplementary Fig. 1b. Alternatively, continuous sampling of *Streptomyces* cultures at different time points for AHL detection to monitor the gene expression timely would be possible (please see newly added Fig. 2d and e). However, VRS-bAHL is particularly useful when the detection of violacein in CV026 is done separately to minimize the interference of secondary metabolites from *Streptomyces*, so using VRS-bAHL for continuous real-time evaluation of gene expression is not the best choice, although it can be done via co-culturing under fine-tuned conditions.

4. The authors should draw the reader's attention to the fact that the mutant they used, M1146, has 4 polyketide clusters removed but leaves the GBL butanolide system and the SARP regulator CpkN intact. Recently, it has been shown that the two elements are related and are important in the global regulatory system. The authors should make sure that there is no interference between the butanolide system and their reporter system. At the very least, this needs to be commented in the text.

Response: Thanks for reasonable concerns. Intriguingly, the results showed that there was no detectable interference of butanolide system on VRS-bAHL (e.g. assays with *S. coelicolor* A3(2) in Fig. 1). Additional experiment with M1146 was performed, and similar results as *S. coelicolor* A3(2) were obtained (see Response Fig. 1 below). Other related comments could also be found in Discussion (P22, lines 434-441).

Response Fig. 1. Feasibility of VRS-bAHL in M1146. M1146 and the derivative strains were grown on MS agar medium and double-layer plate method was applied to detect AHL. We could see that M1146, M1146NC (M1146 containing plasmid pSET152) could not induce violacein production, rendering a clear background, whereas M1146OEcvil, M1146 containing the plasmid pSP_{hrdB-cviI}

(pSET152:: $P_{hrdB-cviI}$), induced violacein production clearly, suggesting that there was no detectable interference between M1146 butanolide system and VRS-bAHL reporter system under present conditions.

REVIEWERS' COMMENTS:

Reviewer #1 (Remarks to the Author):

no comments

Reviewer #3 (Remarks to the Author):

The authors, carefully answered the questions and discussed the issues I pointed out in the review. They have also performed additional experiments that greatly enhance the value of the work. Thank you for the effort made. I am satisfied with the current form of the manuscript. As it stands, the work can be published without comment.